# The Global, Regional, and National Uterine Cancer Burden Attributable to High BMI from 1990 to 2019: A Systematic Analysis of the Global Burden of Disease Study 2019

**DOI:** 10.3390/jcm12051874

**Published:** 2023-02-27

**Authors:** Jingchun Liu, Haoyu Wang, Zhi Wang, Wuyue Han, Li Hong

**Affiliations:** Department of Obstetrics and Gynecology, Renmin Hospital of Wuhan University, Wuhan 430064, China

**Keywords:** uterine cancer, high BMI, death, disability-adjusted life year, global burden of disease

## Abstract

Uterine cancer (UC) is the most common gynecologic malignancy, and high body mass index (BMI) is a poor prognostic factor for UC. However, the associated burden has not been fully assessed, which is crucial for women’s health management and the prevention and control of UC. Therefore, we utilized the Global Burden of Disease Study (GBD) 2019 to describe the global, regional, and national UC burden due to high BMI from 1990 to 2019. The data show that globally, women’s high BMI exposure is increasing annually, with most regions having higher rates of high BMI exposure than the global average. In 2019, 36,486 [95% uncertainty interval (UI): 25,131 to 49,165] UC deaths were attributed to high BMI globally, accounting for 39.81% (95% UI: 27.64 to 52.67) of all UC deaths. The age-standardized mortality rate (ASMR) and age-standardized disability-adjusted life years (DALY) rate (ASDR) for high BMI-associated UC remained stable globally from 1990 to 2019, with significant differences across regions. Higher ASDR and ASMR were found in higher socio-demographic index (SDI) regions, and lower SDI regions had the fastest estimated annual percentage changes (EAPCs) for both rates. Among all age groups, the fatal outcome of UC with high BMI occurs most frequently in women over 80 years old.

## 1. Introduction

Uterine cancer (UC) is the most common gynecological malignancy [1]. It is estimated that about 7% of all new cancers in women in the United States are UCs. Although most UCs are highly treatable with a good prognosis [2], the estimated mortality rate is still the sixth highest among female cancers whose disease burden cannot be ignored. Previous studies have estimated differences in the incidence, severity, or mortality of UC by age, race, and ethnicity [3,4,5,6,7]. Assessing the disease burden of UC due to other important risk factors is essential to guide disease interventions and improve public health. High body mass index (BMI)has been shown to be an important risk factor for UC with an association with higher all-cause mortality and disease recurrence [8,9,10]. However, due to the rising trend of high BMI in women worldwide, there is a need to strengthen health policies and management, which will benefit UC control and reduce the risk of the burden of other diseases [11,12].

The Global Burden of Disease Study (GBD) 2019 focuses on the burden of multiple diseases and injuries and incorporates the associated risk factors and the relative harms they cause. The Socio-demographic Index (SDI) is a composite indicator of the development status of countries or regions based on a combination of per capita income, education, and fertility rates [13]. In this study, we used GBD 2019 to provide a comprehensive analysis of the disease burden of UC due to high BMI from global, regional, and national levels, including risk factor exposure, disability-adjusted life years (DALYs), and deaths, with links to SDI, aiming to contribute to women’s health management, as well as prevention and control of UC.

## 2. Method

### 2.1. Data Source

We obtained the data for this study from the GBD 2019 public dataset (http://ghdx.healthdata.org/gbd-results-tool (accessed on 13 October 2022)). GBD 2019 focuses on 369 diseases and injury burdens and incorporates their associated risk factors and the relative harms they cause. We restricted our analysis to deaths and DALYs due to UC in total and attributable to high BMI, as well as summary exposure values (SEVs) of women with high BMI worldwide. Our study was grouped by age (5-year age groups from 20 to 79 years and 80+ years) and region. In GBD 2019, the 204 countries and territories in the study were classified into five levels based on the SDI, a composite indicator that assesses total fertility, education, and income attainment. They are also divided into 21 GBD regions based on geographical location.

The statistical indicators used in this study on age, death, DALY, SEV, etc., are available for free by visiting GBD 2019. The 95% uncertainty interval (UI) of data was obtained by accessing GBD 2019.

### 2.2. Standard Definitions

In GBD 2019, UC is defined as C54–C54.3, C54.8–C54.9, Z85.42, and Z86.001 to match ICD10. Data for high BMI in GBD 2019 were obtained by updating and supplementing the GBD 2017. Criteria for high BMI were defined by mixed-effects modeling and data adjustment according to each country, year, age, and sex. In general, a high BMI for adults (ages 20+) was defined as BMI greater than 25 kg/m^2^ [14,15]. The SDI values divide all countries and regions into five levels, and the exact list can be found in previous studies [15].

### 2.3. Statistical Analysis

Death, the age-standardized mortality rate (ASMR), DALY, and age-standardized DALY rate (ASDR) were used to quantify the burden of uterine cancer attributable to high BMI worldwide. Population attribution fractions (PAFs) are used to assess the burden of disease attributable to a risk factor and are calculated as E/O × 100%, where E is the number of cases that can be attributed to the exposure and O is the total number of cases. Time trends in statistical indicators from 1990 to 2019 are measured by the estimated annual percentage change (EAPC), which is calculated as EAPC = 100 × (exp (β) − 1), with β representing the annual change in ln (age-standardized rate) and the 95% confidence interval (CI) also derived from the model. Positive EAPC here represents an increasing trend, and negative EAPC represents a decreasing trend. Finally, Spearman’s correlation was used to test the association between the statistical indicators, SDI and EAPC. All statistical analysis and data visualization is performed with the software GraphPad Prism8 (Boston, MA, USA) or R (4.1.0) (Auckland, New Zealand). 

### 2.4. Ethics Statement

No ethical review was required for this study because only extensive pooled data without personal identifiers were used in the data analysis.

## 3. Results

### 3.1. UC Deaths and ASMR Attributable to High BMI

From 1990 to 2019, the PAF of UC deaths attributable to high BMI increased from 30.65% (95% UI: 19.42 to 43.94) to 39.81% (95% UI: 27.64 to 52.67) annually worldwide (Appendix A). Specifically, the number of UC deaths associated with high BMI increased from 56,130 (95% UI: 51,104 to 60,199) to 91,641 (95% UI: 82,389 to 101,502), and ASMR was relatively stable within the fluctuating range, developing from 0.82 (95% UI: 0.51 to 1.17) to 0.83 (95% UI. 0.57 to 1.12) (Figure 1A–C). The EAPC was 0.05 (95% CI: 0 to 0.09).

At the SDI-regional level, the number of UC deaths attributable to high BMI and the proportion of total UC deaths has increased annually in all five regions. Deaths due to UC associated with high BMI occurred most frequently in High SDI and High-middle SDI areas, with 11,964 (95% UI: 8392 to 15,777) and 12,148 (95% UI: 8455 to 16,245) deaths in 2019, respectively (Figure 1B and Table 1). From 1990 to 2019, ASMR increased yearly in almost all SDI regions, reaching a range of 0.51–1.15 in 2019, except for High-middle SDI which decreased from 1.22 to 1.07 (Figure 1C). The same trend is reflected in the EAPC, which stands at −0.65 (95% CI: −0.56 to −0.75) in the High-middle SDI and ranges from 0.53 to 2.23 in the other regions.

At the GBD-regional level, high BMI-associated UC deaths occurred mainly in High-income North America, Western Europe, and Eastern Europe (Figure 1D and Table 1). From 1990 to 2019, Central Asia, High-income Asia Pacific, and Eastern Europe showed a decreasing trend in ASMR, with EAPC ranging from −0.41 to −0.37. ASMR volatility was relatively stable or increasing in other regions, with Southeast Asia showing the fastest growth in ASMR (EAPC = 3.16; 95% CI: 3.05 to 3.27).

At the national level, the most deaths due to high BMI in UC were caused by the United States of America in both 1990 and 2019, followed by the Russian Federation and China (Appendix A). The highest ASMR was observed in American Samoa in 1990 (ASMR = 3.94, 95% UI: 2.57 to 6.1) and 2019 (ASMR = 6.06, 95% UI: 3.71 to 8.57) (Figure 2). During this period, Bangladesh had the lowest ASMR of 0.06 (95% UI: 0.01 to 0.18) in 1990 and 0.14 (95% UI: 0.06 to 0.34) in 2019. There were 162 countries with elevated ASMR out of the 204 countries and territories included in the analysis, and Equatorial Guinea showed the fastest increase with EAPC in ASMR of 6.84 (95% CI: 6.11 to 7.56). Among other countries and territories, the Republic of Korea experienced the largest decline with an EAPC in ASMR of −3.44 (95% CI: −4.29 to −2.58).

In 2019, high BMI-related UC deaths were associated with age, peaking in the 80+ age group, followed by the 65–69 age group (Figure 3). Age-specific mortality rate increased with age globally and in all SDI level regions. Specifically, deaths occurred most frequently in the 80+ age group, with the majority occurring in the High SDI and High-middle SDI regions. Globally, the EAPC for age-specific mortality rate developed a W-shaped association with age, with values below 0 at ages 45–49 and 65–79, decreasing fastest at ages 60–69 and increasing fastest at ages 80+. At the SDI regional level, the EAPC for age-specific mortality rate in the High-middle SDI region was chronically below 0 until 2019. All other regions have positive values, with persistently higher values in the Low-middle SDI and Low SDI regions.

### 3.2. UC DLAYs and ASDR Attributable to High BMI

From 1990 to 2019, the PAF of UC DALY cases attributable to high BMI increased from 30.00% to 40.19% worldwide (Appendix A). The number of high BMI-associated UC DALY cases increased from 444,333 (95% UI: 276,290 to 633,114) to 935,961 (95% UI: 642,880 to 1,255,462) (Figure 4A, B). The ASDR increased from 20.59 (95% UI: 12.82 to 29.38) in 1990 to 21.48 (95% UI: 14.75 to 28.83) in 2019, with an EAPC of 0.16 (95% CI: 0.11 to 0.21) (Table 1 and Figure 4C).

At the SDI-regional level, the number and proportion of UC DALYs attributed to high HBI increased year on year for all five levels of SDI regions. In 2019, the high-middle SDI region contributed the most DALYs for high BMI-associated UC, although the corresponding ASDR showed a decreasing trend over three decades with an EAPC of −0.71 (95% CI: −0.82 to −0.6). All other regions showed an increasing trend in ASDR, and in particular, the fastest-growing ASDR was observed in the Low SDI region (EAPC = 2.24; 95% CI: 2.15 to 2.34) (Figure 4C).

Among all 21 GBD regions, the top three regions with the highest occurrence of UC DALY attributable to high BMI were High-income North America, Western Europe, and Eastern Europe (Figure 4D). In 1990, the highest ASDR belonged to Eastern Europe (ASDR = 59.35; 95%UI: 40.42 to 78.12), while in 2019, it was the Caribbean (ASDR = 65.75; 95% UI: 44.11 to 90.18). From 1990 to 2019, the ASDR for Eastern Europe and Central Asia declined, with EAPCs of −0.42 (95% CI: −0.68 to −0.15) and −0.41 (95% CI: −0.58 to −0.25), respectively. The ASDR for Southern Latin America, Andean Latin America, and High-income Asia Pacific is relatively stable. Other regions are on an upward trend, with Southeast Asia showing the fastest growth with an EAPC in ASDR of 3.14 (95% CI: 2.97 to 3.3).

At the national level, in 1990, the most DALY cases were found in Russian Federation, and in 2019 were contributed by the United States of America (Figure 5 and Appendix A). During the period, American Samoa and Bangladesh consistently occupied the highest and lowest ASDRs, with 108.2 (95% UI: 71.33 to 164.82) and 1.65 (95% UI: 0.26 to 4.76) in 1990, 3.83 (95% UI: 1.55 to 9.10) and 171.18 (95% UI: 106.68 to 237.06) in 2019, respectively. From 1990 to 2019, a total of 45 countries and territories had declining ASDRs, and 159 were stable or increasing. The Republic of Korea had the most significant ASDR decline (EAPC = −3.38; 95% CI: −4.35 to −2.4), and Equatorial Guinea had the fastest ASDR increase (EAPC = 6.32; 95% CI: 5.58 to 7.06).

In 2019, the number of age-specific DALYs peaked at age 60–64 years globally, yet age-specific DALY rates showed a similar pattern to death, peaking at age 65–69 years (Figure 6). The age-specific DALY numbers and rates for all SDI regions were similar to the global trend with inverted V patterns. In addition, the EAPC in age-specific DALY rates showed similar trends to the EAPC in age-specific mortality rates.

### 3.3. The Changes in High BMI Exposure in Women

The age-standardized SEV rate of high BMI among women increased globally, from 12.44 (95% UI: 9.05 to 17.05) in 1990 to 20.59 (95% UI: 16.66 to 25.75) per 100,000 persons in 2019. All SDI regions are in line with global average trends, while the high SDI region consistently shows the highest age-standardized SEV rate of high BMI (Figure 7). There is a total of 13 GBD regions with age-standardized SEV rates of high BMI among women consistently above the global average over the three decades, particularly in high-income North America and Southern Sub-Saharan Africa (Appendix A).

### 3.4. Association of ASMR, ASDR of UC Attributable to High BMI with SDI

A cross-sectional analysis between ASMR of high BMI-associated UC and SDI over 30 years in the GBD regions revealed that ASMR first increased slowly with SDI until it reached approximately 0.5, then increased rapidly, and decreased rapidly after SDI reached 0.7 (Figure 8A). Overall, in 2019, EAPC in ASMR of UC attributable to high BMI was negatively associated with SDI values (R = −0.506, *p* = 0.012), reaching the highest EAPC at approximately SDI of 0.53 and the lowest at 0.75 (Figure 8C). In the GBD region, the association between ASDR or its EAPC and SDI has a similar pattern to that of ASMR (Figure 8B,D).

## 4. Discussion

Previous studies have described the overall burden of uterine cancer over 30 years [16]. In fact, excess body fat and high BMI have been implicated as important causes of most cancers [17,18]. Multiple studies have reported an association between increased BMI and UC risk [12,19,20,21,22] and a stronger association between BMI and risk of endometrioid adenocarcinoma [23,24]. Current genome-wide association studies (GWAS) suggest that high BMI is one of the most definitive risk factors for endometrial cancer and identified risk loci and basis for the risk stratification [25]. Additionally, studies have shown that obesity affects the mapping of dye to the sentinel lymph node in minimally invasive procedures suggesting an intervention for treatment [26]. Mechanistically, obesity is associated with high levels of circulating estrogens in the body, abnormal fatty acid metabolism, and long-term chronic inflammation of the microenvironment, which may promote the development of cancer cells [27,28,29,30]. Therefore, understanding the burden of UC attributable to high BMI and providing timely health surveillance and disease control for risk populations to reduce the disease burden on individuals and society is critical to the current prevention and control of UC worldwide.

In this study, spatial and temporal trends in mortality and DALY attributable to high BMI in UC were estimated at the global, regional, and national levels. A series of analyses have shown that the global burden of UC attributable to high BMI is large and growing. From 1990 to 2019, the number of UC deaths associated with high BMI almost doubled, with the percentage increasing from 30.65% to 39.81%. Meanwhile, the percentage of DALYs attributable to high BMI increased from 30.00% to 40.19%. Moreover, over the past three decades, while high-middle SDI areas have the highest burden of high BMI-associated UC, low SDI areas show faster-increasing mortality and DALY rates. This is consistent with previous reports [31] and may reflect ongoing epidemiologic shifts, demographic changes, and disparities in UC prevention and health control. In conclusion, these results provide a more comprehensive and comparable estimate that may inform a fair and reasonable reduction in the global burden of UC, particularly high BMI-associated UC.

Although the absolute burden of high BMI-associated UC increased from 1990 to 2019, the global ASMR (EAPC = 0.05, 95% CI: 0 to 0.09) and ASDR (EAPC = 0.16, 95%, CI:0.11 to 0.21) were relatively stable. The ASMR and ASDR due to high BMI-associated UC decreased or remained relatively stable despite higher high BMI exposure in areas with higher SDI. Notably, however, these two rates remained significantly higher in areas with lower SDI. In addition, high BMI exposure and the associated disease burden of UC varied widely worldwide; high-income North America, Western Europe, and Eastern Europe consistently accounted for the highest-burden over three decades. At the national level, Equatorial Guinea, Uganda, and Lesotho showed a rapid increase in ASDR and ASMR. These results suggest that some progress may have been made in the control of high BMI-associated UC globally over the past three decades, and they are cautiously optimistic. However, the uneven distribution of the burden of high BMI-associated UC and disparities in increase around the world diminish the potential to progress and suggest the need to adapt UC control efforts to specific societal circumstances and health system needs, taking into account disease context, regional culture, and high BMI exposure, to accelerate efforts to control and address the burden of high BMI-associated UC and regional inequalities.

From 1990 to 2019, the highest and fastest-growing rates of high BMI-associated UC mortality were concentrated in people over 80 years of age, probably due to the difficulty of treatment and health control caused by the decline in the body’s general condition with increasing age. Meanwhile, cancer control and care in older adults is complex, with a potentially blurred demarcation between reasonable treatment, over-treatment and under-treatment [32]. The highest DALY rates were concentrated among 65–69-years-olds adults, but the fastest increases were found among 25–29 and 55–59-year groups. This implies that the trend toward a younger burden of uterine cancer associated with high BMI is also a challenge that cannot be ignored. Previous studies have shown that low alcohol, reasonable physical activity, and having a college degree are associated with a healthy BMI in young women [33]. However, poor lifestyle habits, occupational stress, discordant emotional states, and major illnesses may lead to high BMI. All these suggest that reasonable guidance and improvement of life and health status will be meaningful in the prevention and treatment of uterine cancer in young women. The Research Aimed at Improving Both Mood and Weight (RAINBOW), which co-treats mood and obesity and has shown benefits in multiple trials, maybe a suitable model to follow [34,35]. Higher SDI regions still lead to age-specific high BMI-associated UC mortality and DALY rates. These data remind us that the burden of disease cannot be controlled in a one-size-fits-all idealistic manner, and the age of the population and the risks it may carry a need to be taken into account. GBD 2019 builds on the changing global cancer burden landscape and demonstrates the importance of social environment, population age, risk context, and other characteristics of cancer risk, which can be helpful for directions and strategies for disease risk control.

Several limitations provide opportunities for future improvements to this GBD 2019-based study. First, the definition of GBD data needs to be clarified. For UC, data on nonmalignant tumors of the cervix in situ (ICD10: Z86.001) were included. This classification does not seem appropriate and is highly controversial, as it is usually assessed as precancerous of the cervix malignancy rather than UC, based on the location and pathological features of the lesion. Second, data for UC can be refined because it is a broad concept that includes many diseases. Providing more specific burden data for disease classification (e.g., endometrial cancer), clinical classification, and even molecular typing would help in further public health management. Third, the criteria for high BMI were defined as greater than 25 kg/m^2^ according to country, sex, age, and year, with 20 to 25 kg/m^2^ being used as the minimum detection limit. However, this crude assessment of results should be improved, and specific criteria should be listed to increase credibility and facilitate individual analysis. Another limitation that cannot be ignored is the lack of available or high-quality data in some areas, resulting in lower data accuracy and delayed availability, which may overestimate or underestimate the burden of disease in a given area or time. Differences in data availability and reliability among SDI regions, GBD regions, and countries can lead to wide variations in trends in disease burden estimates. Therefore, increased global normative surveillance of cancer and health is needed. The introduction and dissemination of new technologies or methods may help identify more women at higher risk of disease and establish screening [36]. In addition, this study, which continued through 2019, did not consider the association between changes in social environment or health management and UC attributable to high BMI during the COVID-19 pandemic, which may have a significant impact on the global pattern of disease burden. Assessment of these associations is critical for future work on the burden of UC because a potential lack of health control, poor diagnosis, and limited treatment may be detrimental to global efforts to reduce cancer burden [37,38,39,40].

In conclusion, a systematic analysis of UC attributable to high BMI in the GBD 2019 study provides comprehensive and comparable estimates of the burden of this disease. The burden of high BMI-associated UC is not negligible, with the growth of high BMI exposure in most regions globally over the past three decades. These estimates vary considerably worldwide, highlighting the inequality in the global burden of UC. This study demonstrates the need to enhance health surveillance and disease control with precision based on characteristics such as social environment, population age, and risk background to reduce the impact of the growing burden of UC.

## Figures and Tables

**Figure 1 jcm-12-01874-f001:**
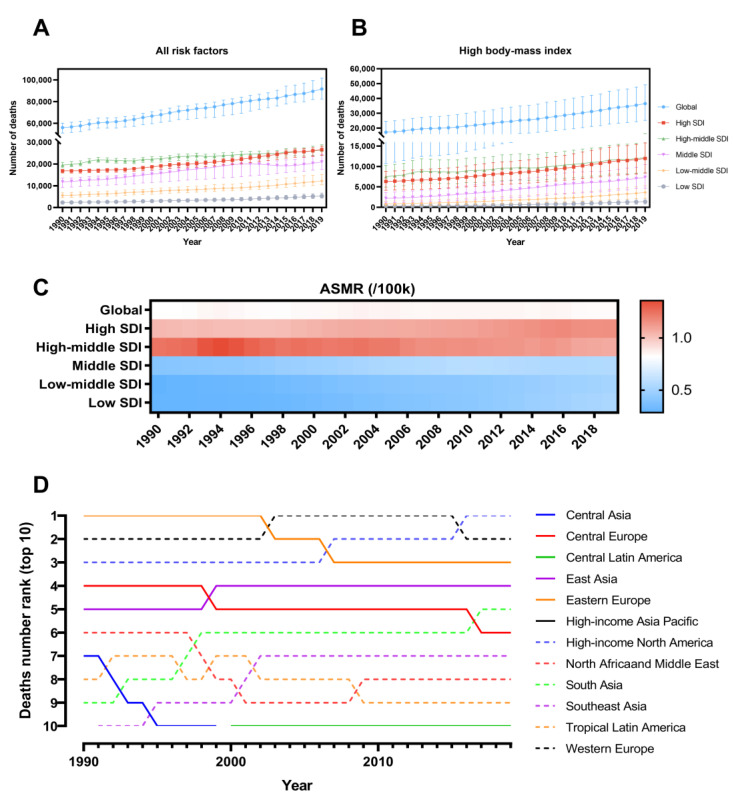
(**A**) Number of uterine cancer deaths under all risk factors from 1990 to 2019. (**B**) Number of uterine cancer deaths attributable to high BMI. (**C**) ASMR from 1990 to 2019 for global and SDI regions. (**D**) Ranking of uterine cancer deaths attributable to high BMI in GBD regions, 1990–2019 (Top 10). BMI, Body mass index; ASMR, Age-standardized mortality rate; SDI, Socio-demographic Index; GBD, Global Burden of Disease Study.

**Figure 2 jcm-12-01874-f002:**
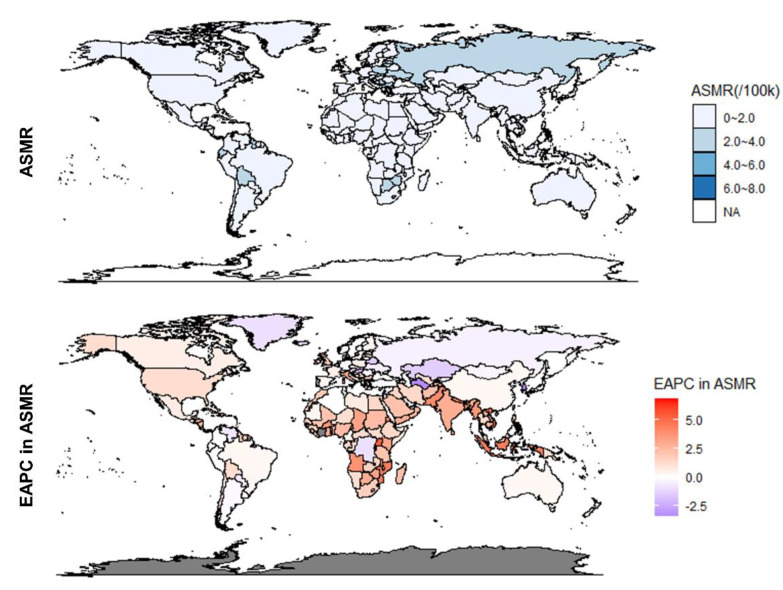
ASMR in 2019 for 204 countries and regions and their EAPC between 1990 and 2019. ASMR, Age-standardized mortality rate; EAPC, Estimated annual percentage change.

**Figure 3 jcm-12-01874-f003:**
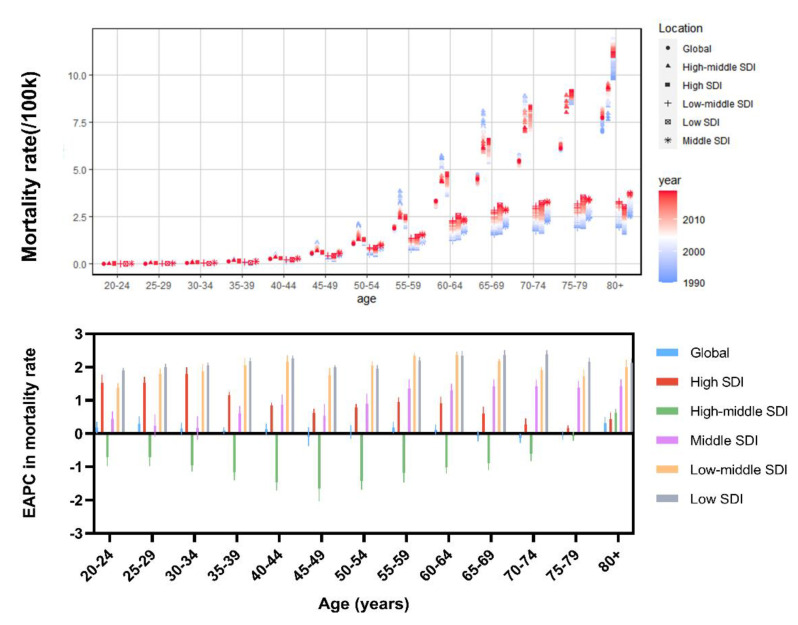
Mortality rate among different age groups in the global and SDI regions in 2019 and their EAPC between 1990 and 2019. SDI, Socio-demographic Index; EAPC, Estimated annual percentage change.

**Figure 4 jcm-12-01874-f004:**
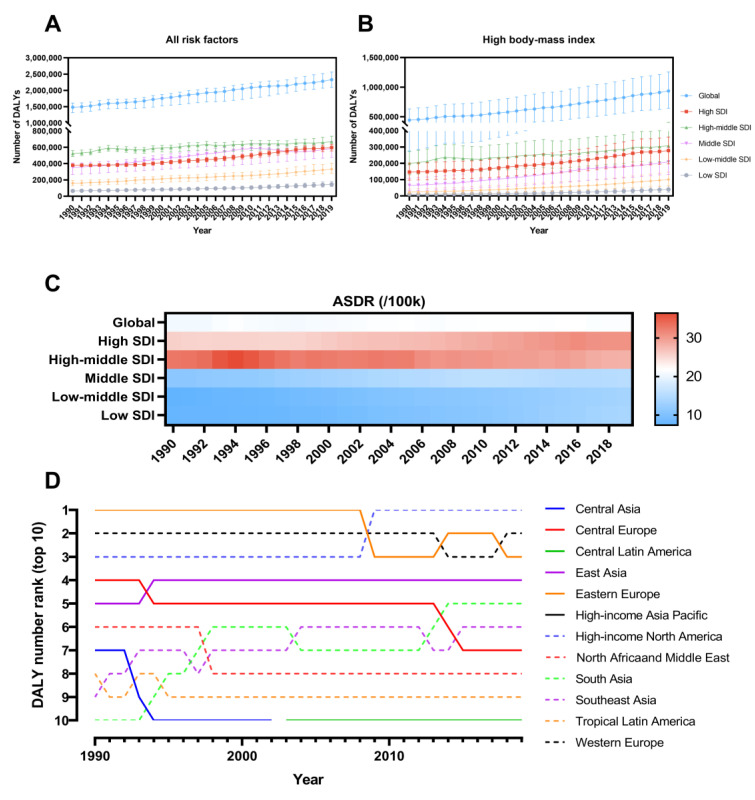
(**A**) Number of uterine cancer DALYs under all risk factors from 1990 to 2019. (**B**) Number of uterine cancer DALYs attributable to high BMI. (**C**) ASDR from 1990 to 2019 for global and SDI regions. (**D**) Ranking of uterine cancer DALYs attributable to high BMI in GBD regions, 1990–2019 (Top 1). DALY, Disability-adjusted life year; ASDR, Age-standardized DALY rate; SDI, Socio-demographic Index; GBD, Global Burden of Disease Study.

**Figure 5 jcm-12-01874-f005:**
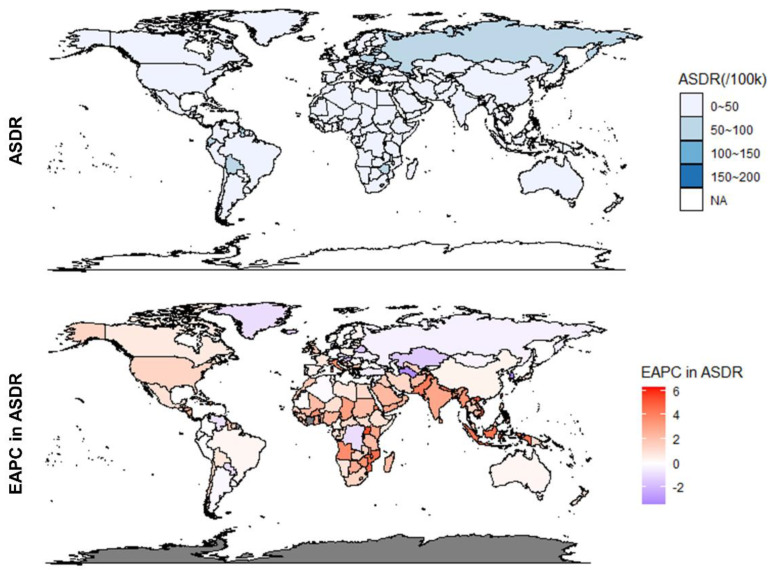
ASDR in 2019 for 204 countries and regions and their EAPC between 1990 and 2019. ASDR, Age-standardized DALY rate; EAPC, Estimated annual percentage change.

**Figure 6 jcm-12-01874-f006:**
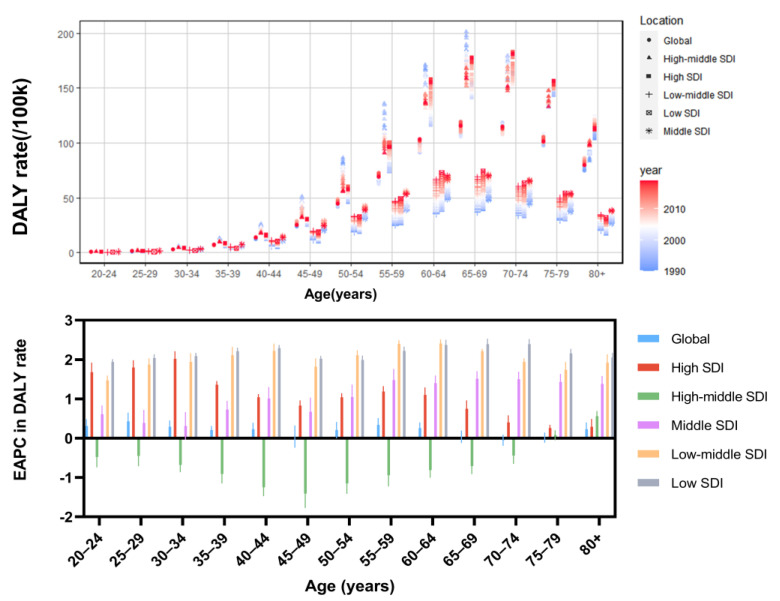
Mortality rate among different age groups in the global and SDI regions in 2019 and their EAPC between 1990 and 2019. SDI, Socio-demographic Index; EAPC, Estimated annual percentage change.

**Figure 7 jcm-12-01874-f007:**
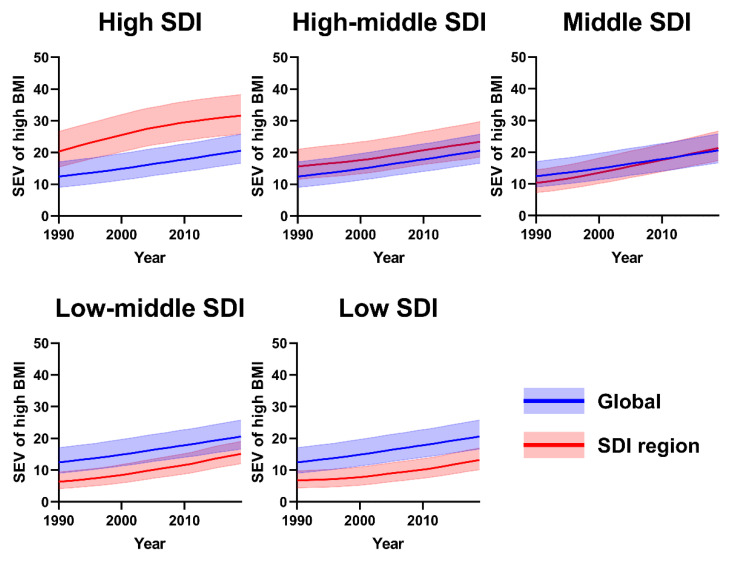
SEV of high BMI for women (per 100) in global and SDI regions from 1990 to 2019. Red represents SDI regions, and blue represents global levels. SEV, Summary exposure values; SDI, Socio-demographic Index.

**Figure 8 jcm-12-01874-f008:**
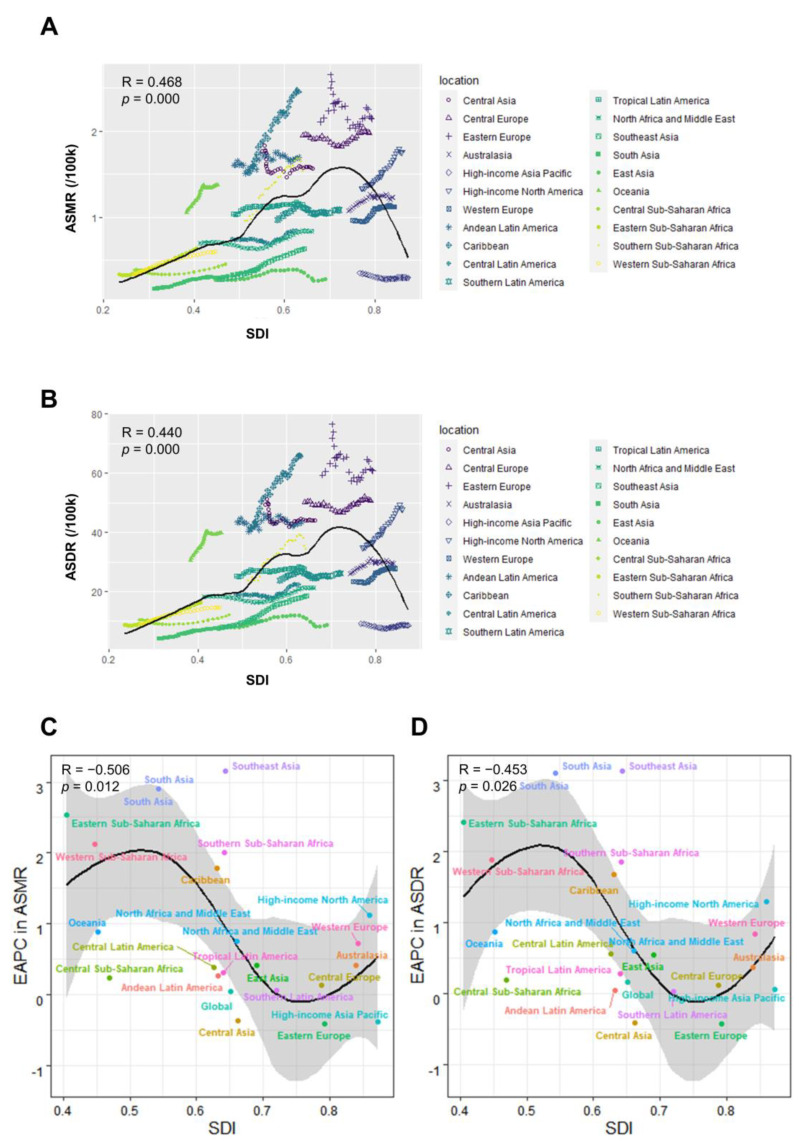
(**A**) Association of SDI levels with ASMR for uterine cancer attributable to high BMI in GBD regions from 1990 to 2019. (**B**) Association of SDI levels with ASDR for uterine cancer attributable to high BMI in GBD regions from 1990 to 2019. (**C**) Association of the GBD regional SDI with EAPC in ASMR in 2019. (**D**) Association of the GBD regional SDI with EAPC in ASDR in 2019. SDI, Socio-demographic Index; ASMR, Age-standardized mortality rate; ASDR, Age-standardized DALY rate; EAPC, Estimated annual percentage change.

**Table 1 jcm-12-01874-t001:** Burden of uterine cancer attributable to high BMI in global, SDI regions, and GBD regions.

Characteristics	1990				2019				1990–2019	
	Deaths Cases	ASMR (/100 k)	DALYs	ASDR (/100 k)	Deaths Cases	ASMR (/100 k)	DALYs	ASDR (/100 k)	EAPC in ASMR	EAPC in ASDR
	No. (95% UI)	No. (95% UI)	No. (95% UI)	No. (95% UI)	No. (95% UI)	No. (95% UI)	No. (95% UI)	No. (95% UI)	No. (95% CI)	No. (95% CI)
Global	17,189 (10,716 to 24,552)	0.82 (0.51 to 1.17)	444,333 (276,290 to 633,114)	20.59 (12.82 to 29.38)	36,486 (25,131 to 49,165)	0.83 (0.57 to 1.12)	935,961 (642,880 to 1,255,462)	21.48 (14.75 to 28.83)	0.05 (0 to 0.09)	0.16 (0.11 to 0.21)
SDI region										
High SDI	6343 (4161 to 8803)	1.03 (0.68 to 1.43)	146,375 (96,900 to 202,687)	25.64 (17.02 to 35.45)	11,964 (8392 to 15,777)	1.15 (0.82 to 1.51)	277,948 (196,534 to 361,673)	30.59 (21.74 to 39.67)	0.53 (0.47 to 0.59)	0.81 (0.73 to 0.89)
High-middle SDI	7497 (4941 to 10,266)	1.23 (0.81 to 1.69)	199,595 (130,069 to 272,533)	32.8 (21.39 to 44.92)	12,148 (8455 to 16,245)	1.07 (0.75 to 1.43)	307,750 (213,434 to 407,394)	28.13 (19.47 to 37.4)	−0.65 (−0.75 to −0.56)	−0.71 (−0.82 to −0.6)
Middle SDI	2165 (1083 to 3636)	0.41 (0.2 to 0.68)	64,347 (31,997 to 108,978)	11.11 (5.56 to 18.78)	7326 (4627 to 10,444)	0.56 (0.35 to 0.8)	208,902 (131,242 to 296,042)	15.21 (9.56 to 21.58)	1.24 (1.03 to 1.44)	1.24 (1.03 to 1.46)
Low-middle SDI	832 (381 to 1484)	0.29 (0.13 to 0.51)	23,879 (10,963 to 42,615)	7.36 (3.38 to 13.08)	3620 (2194 to 5248)	0.51 (0.31 to 0.74)	100,545 (61,182 to 145,823)	13.42 (8.15 to 19.44)	2.08 (1.99 to 2.16)	2.15 (2.07 to 2.23)
Low SDI	340 (139 to 641)	0.29 (0.12 to 0.56)	9822 (4093 to 18,346)	7.62 (3.15 to 14.32)	1393 (792 to 2179)	0.53 (0.3 to 0.84)	39,878 (22,869 to 61,395)	13.82 (7.88 to 21.4)	2.23 (2.14 to 2.33)	2.24 (2.15 to 2.34)
GBD regions										
Andean Latin America	170 (101 to 249)	1.61 (0.94 to 2.37)	4913 (2961 to 7080)	43.47 (25.99 to 62.77)	492 (310 to 710)	1.69 (1.07 to 2.44)	12,845 (8171 to 18,578)	43.28 (27.55 to 62.7)	0.27 (0.14 to 0.4)	0.05 (−0.09 to 0.18)
Australasia	141 (90 to 200)	1.08 (0.69 to 1.52)	3235 (2089 to 4552)	26.09 (16.86 to 36.54)	332 (232 to 437)	1.23 (0.87 to 1.62)	7198 (5127 to 9384)	29.58 (21.27 to 38.27)	0.42 (0.29 to 0.54)	0.37 (0.24 to 0.51)
Caribbean	206 (132 to 292)	1.53 (0.98 to 2.17)	5762 (3700 to 8066)	41.59 (26.76 to 58.21)	675 (458 to 924)	2.46 (1.67 to 3.36)	17,767 (11,922 to 24,315)	65.75 (44.11 to 90.18)	1.79 (1.67 to 1.92)	1.68 (1.53 to 1.83)
Central Asia	488 (337 to 655)	1.76 (1.22 to 2.37)	13,815 (9571 to 18,491)	49.28 (34.13 to 65.92)	665 (469 to 872)	1.57 (1.11 to 2.06)	20,180 (14,198 to 26,720)	43.95 (31.11 to 58.03)	−0.37 (−0.56 to −0.18)	−0.41 (−0.58 to −0.25)
Central Europe	1653 (1150 to 2187)	1.95 (1.36 to 2.58)	41,831 (29,294 to 55,338)	50.28 (35.12 to 66.69)	2464 (1753 to 3224)	1.98 (1.4 to 2.59)	56,764 (40,041 to 74,952)	50.72 (35.54 to 66.98)	0.14 (0.02 to 0.25)	0.12 (0.01 to 0.24)
Central Latin America	318 (201 to 454)	0.75 (0.47 to 1.07)	8748 (5524 to 12,379)	18.98 (12.03 to 26.89)	1074 (697 to 1504)	0.84 (0.55 to 1.18)	29,040 (18,902 to 40,960)	22.35 (14.51 to 31.54)	0.39 (0.19 to 0.59)	0.57 (0.35 to 0.79)
Central Sub-Saharan Africa	50 (23 to 88)	0.39 (0.17 to 0.71)	1472 (692 to 2637)	10.48 (4.9 to 18.76)	134 (68 to 230)	0.46 (0.23 to 0.78)	3920 (2008 to 6679)	12.11 (6.18 to 20.61)	0.24 (−0.11 to 0.58)	0.2 (−0.15 to 0.55)
East Asia	1260 (328 to 2694)	0.27 (0.07 to 0.58)	39,517 (10,137 to 85,016)	8.01 (2.07 to 17.22)	3109 (1374 to 5696)	0.28 (0.12 to 0.51)	95,239 (43,288 to 173,477)	8.66 (3.89 to 15.78)	0.41 (−0.2 to 1.04)	0.55 (−0.08 to 1.17)
Eastern Europe	3825 (2622 to 5042)	2.11 (1.44 to 2.78)	103,208 (70,784 to 135,624)	59.35 (40.42 to 78.12)	4619 (3229 to 6075)	2.15 (1.51 to 2.82)	121,184 (86,293 to 158,947)	60.77 (43.42 to 79.85)	−0.41 (−0.65 to −0.16)	−0.42 (−0.68 to −0.15)
Eastern Sub-Saharan Africa	127 (51 to 244)	0.33 (0.13 to 0.65)	3705 (1502 to 7108)	8.76 (3.54 to 16.9)	534 (296 to 828)	0.64 (0.35 to 1)	15,194 (8626 to 23,441)	16.28 (9.18 to 25.15)	2.53 (2.29 to 2.78)	2.42 (2.18 to 2.66)
High-income Asia Pacific	399 (152 to 710)	0.35 (0.13 to 0.63)	10,370 (3993 to 18,416)	9.23 (3.55 to 16.43)	673 (284 to 1155)	0.3 (0.13 to 0.5)	15,573 (6814 to 26,306)	8.58 (3.71 to 14.46)	−0.38 (−0.58 to −0.18)	0.07 (−0.17 to 0.31)
High-income North America	2752 (1831 to 3726)	1.33 (0.89 to 1.79)	65,766 (44,759 to 86,605)	34.83 (23.8 to 45.55)	6065 (4290 to 7641)	1.76 (1.25 to 2.2)	151,362 (108,620 to 190,162)	47.84 (34.76 to 60)	1.12 (1.04 to 1.2)	1.3 (1.22 to 1.38)
North Africa and Middle East	588 (364 to 845)	0.7 (0.43 to 1)	16,841 (10,414 to 24,182)	18.16 (11.24 to 25.98)	1728 (1093 to 2295)	0.84 (0.53 to 1.12)	48,564 (30,208 to 65,024)	21.24 (13.26 to 28.45)	0.75 (0.45 to 1.04)	0.61 (0.32 to 0.9)
Oceania	16 (9 to 26)	1.05 (0.55 to 1.68)	528 (288 to 831)	30.42 (16.34 to 47.84)	51 (26 to 81)	1.37 (0.68 to 2.22)	1659 (829 to 2674)	39.79 (20.11 to 64.15)	0.88 (0.73 to 1.03)	0.87 (0.7 to 1.04)
South Asia	430 (153 to 863)	0.17 (0.06 to 0.34)	12,095 (4374 to 24,178)	4.12 (1.47 to 8.25)	2746 (1564 to 4080)	0.39 (0.22 to 0.58)	74,669 (42,897 to 110,340)	9.93 (5.69 to 14.68)	2.9 (2.68 to 3.13)	3.1 (2.92 to 3.29)
Southeast Asia	395 (144 to 749)	0.27 (0.1 to 0.53)	12,602 (4743 to 23,799)	8.1 (3 to 15.32)	2173 (1187 to 3261)	0.63 (0.35 to 0.95)	66,772 (35,238 to 99,996)	18.57 (9.92 to 27.75)	3.16 (3.05 to 3.27)	3.14 (2.97 to 3.3)
Southern Latin America	250 (143 to 371)	0.97 (0.55 to 1.44)	6110 (3496 to 9064)	23.83 (13.64 to 35.36)	513 (344 to 700)	1.08 (0.73 to 1.47)	11,653 (7813 to 15,824)	26.07 (17.51 to 35.34)	0.06 (−0.1 to 0.21)	0.03 (−0.09 to 0.16)
Southern Sub-Saharan Africa	146 (99 to 199)	0.97 (0.65 to 1.31)	3804 (2568 to 5071)	23.62 (16.15 to 31.62)	470 (306 to 614)	1.49 (0.97 to 1.95)	11,490 (7659 to 15,117)	34.5 (22.79 to 45.45)	2 (1.71 to 2.28)	1.86 (1.58 to 2.13)
Tropical Latin America	487 (294 to 712)	1.04 (0.62 to 1.52)	12,720 (7813 to 18,434)	25.18 (15.32 to 36.5)	1523 (1078 to 2027)	1.15 (0.81 to 1.52)	37,050 (26,362 to 49,217)	27.66 (19.68 to 36.73)	0.31 (0.24 to 0.39)	0.29 (0.22 to 0.36)
Western Europe	3345 (2156 to 4686)	0.96 (0.62 to 1.35)	73,414 (47,507 to 102,791)	23.27 (15.05 to 32.61)	5879 (3981 to 8016)	1.12 (0.76 to 1.52)	122,379 (84,072 to 165,733)	27.74 (19.1 to 37.39)	0.73 (0.61 to 0.85)	0.85 (0.75 to 0.95)
Western Sub-Saharan Africa	142 (71 to 242)	0.33 (0.17 to 0.57)	3876 (1984 to 6504)	8.63 (4.38 to 14.52)	567 (356 to 830)	0.6 (0.37 to 0.89)	15,458 (9725 to 22,671)	14.69 (9.21 to 21.47)	2.12 (2 to 2.23)	1.89 (1.77 to 2.01)

BMI, Body mass index; SDI, Socio-demographic Index; GBD, Global burden of disease study; ASMR, Age-standardized mortality rate; DALYs, Disability-adjusted life years; ASDR, Age-standardized DALY rate; EAPC, Estimated annual percentage changes; UI, Uncertainty interval.

## Data Availability

All data used in this study can be freely accessed at the GBD 2019 portal (http://ghdx.healthdata.org/gbd-2019 (accessed on 13 October 2022). The code of interest can be provided by contacting the appropriate author.

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
