# Peer review of "The Global, Regional, and National Uterine Cancer Burden Attributable to High BMI from 1990 to 2019: A Systematic Analysis of the Global Burden of Disease Study 2019"

_jcm, 2023, doi:10.3390/jcm12051874_

Round 1

Reviewer 1 Report

The authors analyzed the data of Global Burden of Disease Study 2019 to provide a comprehensive analysis of the disease burden of uterine cancer due to high BMI. There have been analyzes related to cervical cancer or endometrial cancer using the existing Global Burden of Disease Study data, and this study was conducted in a similar way to those studies. Therefore, there is no disagreement about the contents of the analysis, but a more strict explanation of the basic inclusion criteria of the study and the criteria for determining BMI seems to be required.

# Abstract

The abbreviations would be used with explanation (e.g. ASMR, ASDR, SDI, and EAPC)

# P2

In GBD2019, UC is defined as C54-C54.3, C54.8-C54.9, Z85.42 and Z86.001 to match ICD10.

The criteria of uterine cancer should be strictly defined. According to ICD 10 classification, Z86.001 means “Personal history of in-situ neoplasm of cervix uteri”. In-situ neoplasm of cervix uteri would not be included in the criteria of uterine cancer. Medically, it is common to analyze malignant neoplasm of corpus uteri (C54) and malignant neoplasm of cervix uteri separately. I wonder why only in-situ neoplasm of cervix uteri was included in the uterine cancer analysis.

# P2

Death, age-standardized mortality rate (ASMR), DALY and age-standardized DALY rate (ASDR) were used to quantify the burden of breast cancer attributable to high BMI worldwide.

Breast cancer is not the concern of this article.

# It is necessary to explain how the Global Burden of Disease Study classified the criterion of high BMI. It is known that there are differences in the standard of BMI and the effect of BMI on health depending on Eastern and Western countries. I wonder how the researchers reflected these differences in their analysis.

Reviewer 2 Report

In abstract, there are no instruction about ASMR, ASDR, SDI and EAPCs.

I think it's better to put a space between the number and the %.

P2 line 2. 

GBD 2019 public dataseto → GBD 2019 public dataset

GBD2019 or GBD 2019? The space is needed or do you not let me in? Please alline the notation.

In table1, font of characteristics was not allined.  (global, SDI region, etc are “Gothic” and others are “Times New Roman”)

The description of license from Line 196 to 206 has almost the same sentence in following page, is it correct?

URL: https://www.medrxiv.org/content/10.1101/2021.07.19.21260791v1.full

In figure legend, it is better for you to show the information of abbreviations.

It is difficult for the reader to understand the table unless it is divided into several parts.

I believe the following paper is very similar to this article, but is not listed in the references.

https://onlinelibrary.wiley.com/doi/10.1002/cam4.4608

Reviewer 3 Report

The present paper is good and interesting. 

The scientific and clinical impact is acceptable.

However, in my opinion a minor revision is required. The written English is clear but a minor check it would be useful.

Specifically, I suggest authors to add a comment about the increase of uterine cancer among the young women. In addition, new molecular approach of EC (Gwas classification, rainbow trial)

Round 2

Reviewer 1 Report

I thank the authors for their efforts for revision, though the authors' answers still contain a significant amount of ambiguity.

Indicators that contain this ambiguity are key indicators of this study, and I think a more specific consideration on these points should be included in the discussion, and if there is a limitation, it should also be specifically mentioned.

# P2

In GBD2019, UC is defined as C54-C54.3, C54.8-C54.9, Z85.42 and Z86.001 to match ICD10.

The criteria of uterine cancer should be strictly defined. According to ICD 10 classification, Z86.001 means “Personal history of in-situ neoplasm of cervix uteri”. In-situ neoplasm of cervix uteri would not be included in the criteria of uterine cancer. Medically, it is common to analyze malignant neoplasm of corpus uteri (C54) and malignant neoplasm of cervix uteri separately. I wonder why only in-situ neoplasm of cervix uteri was included in the uterine cancer analysis.

Response 3:

Thank you for your detailed review! In fact, according to the ICD10 classification, Z86.0 is defined as ‘Personal history of in-situ and benign neoplasms and neoplasms of uncertain behavior (Excludes: personal history of malignant neoplasms)’. This means that the "Personal history of in-situ neoplasm of cervix uteri" in Z86.001 represents benign neoplasm in situ of the cervix, not malignant neoplasm of the cervix. In GBD 2019, the official body classifies it as a broad category of uterine cancer. We think this is understandable.

As noted by the authors, Z86.0 is defined as ‘Personal history of in-situ and benign neoplasms and neoplasms of uncertain behavior (Excludes: personal history of malignant neoplasms).
By this definition, it is much clear that Z86.001 does not involve malignant disease. The subject of this study is the uterine cancer burden, so a consideration and discussion on whether it is appropriate to include "personal history of in-situ neoplasm of cervix uteri" should be included in this study. If precancerous lesion for endometrial cancer is included, endometrial hyperplasia should also be included. This classification by the official body of the original study does not guarantee the appropriateness of the classification.

Point 5:

# It is necessary to explain how the Global Burden of Disease Study classified the criterion of high BMI. It is known that there are differences in the standard of BMI and the effect of BMI on health depending on Eastern and Western countries. I wonder how the researchers reflected these differences in their analysis.

Response 5:

Thank you for your meaningful suggestions. We agree that a clear definition of high BMI is critical. In fact, the GBD defines high BMI as follows.

High body-mass index (BMI) for adults (ages 20+) is defined as BMI greater than 20 to 25 kg/m2. High BMI for children (ages 1-19) is defined as being overweight or obese based on International Obesity Task Force.

Our study focused on the burden of uterine cancer in adults (ages 20+), so we have added relevant information in the methods section.

Defining high BMI as “greater than 20 to 25 kg/m2” is somewhat confusing. With such an ambiguous expression, it is not known whether a BMI of 21, which is generally considered normal, is included in high BMI.

According to the Table 1 of the reference article no. 14(Global, regional, and national comparative risk assessment of 84 behavioural, environmental and occupational, and metabolic risks or clusters of risks for 195 countries and territories, 1990–2017: a systematic analysis for the Global Burden of Disease Study 2017), the “Theoretical minimum risk exposure level” is suggested as 20–25 kg/m².

If women with a BMI in the 20-23 are also classified as high BMI, consideration of whether this distinction is appropriate should be included in the text.
